# Health risk communication and infodemic management in Iran: development and validation of a conceptual framework

Azam Bazrafshan,[1] Azadeh Sadeghi,[1,2] Maliheh Sadat Bazrafshan,[1] Hossein Mirzaie,[1] Mehdi Shafiee,[1,2] Jaason Geerts  ,[3,4] Hamid Sharifi  [1]

[1]HIV/STI Surveillance Research Center, and WHO Collaborative Center for HIV Surveillance, Institute for Futures Studies in Health, Kerman University of Medical Sciences, Kerman, Iran (the Islamic Republic of)
[2]Deputy of Health, Department of Communicable Diseases, Kerman University of Medical Sciences, Kerman, Iran (the Islamic Republic of)
[3]Research and Leadership Development, Canadian College of Health Leaders, Ottawa, Ontario, Canada
[4]Bayes Business School, University of London, London, UK

**Correspondence to**
Dr Hamid Sharifi;
sharifihami@gmail.com

## ABSTRACT

**Objective** The COVID-19 pandemic exposed significant gaps in Iran's and other health systems' risk communication. The accompanying infodemic undermined policy responses, amplified distrust in government and reduced adherence to public health recommendations among the Iranian population. This study aimed to develop a conceptual framework for health risk communication and infodemic management (RCIM) during epidemics and health emergencies in Iran that could have potential applications in other contexts.

**Design** This study was designed in two phases. Phase 1 involved semistructured qualitative interviews with key informants to explore effective RCIM strategies across public health settings in Iran and to develop a conceptual framework. Phase 2 involved revising the framework based on feedback from an online expert panel regarding its comprehensiveness and validity.

**Setting** Provincial/national public health settings in Iran.

**Participants** Twenty key informants from provincial and national public health authorities who contributed to COVID-19 response programmes participated in interviews. Nine experts from diverse academic disciplines, provincial and national settings, and geographical locations participated in an online expert panel.

**Results** The conceptual model was created based on qualitative interviews and expert panel discussions and was structured according to six pillars of the WHO health system framework: leadership and governance, information, health workforce and financial resources, along with media and community. Leadership and governance, including trustworthy leaders, were recommended as the foundation for developing RCIM in Iran. Developing an official strategy with information infrastructures, including high-quality surveillance systems, identified personnel and training for specialists among the health workforce, financial resources, communication channels and community engagement were recognised as other dimensions for developing health risk communication in Iran.

**Conclusion** The proposed framework represents a step toward establishing a national RCIM strategy in Iran. Further validation of the conceptual framework and experiments on how it could potentially influence policy and practice is recommended. This model has the potential to be applied in other contexts in its current form or as the foundation for customised local versions.

## STRENGTHS AND LIMITATIONS OF THIS STUDY

⇒ This study consolidates insights from the field experiences of public health professionals across provincial and national settings in risk communication and infodemic management (RCIM) during the COVID-19 pandemic in Iran. Investigating the experiences and perceptions of academics, health professionals and policymakers enhances the validity of the results by including diverse perspectives on the topic of RCIM, and strengthens the proposed framework's credibility by providing a comprehensive understanding of its applicability in provincial and national public health settings.

⇒ This study presents a novel conceptual framework, validated through full consensus by a panel of experts, for RCIM during epidemics and health emergencies in Iran.

⇒ The qualitative nature of our study and the focus on one country may limit the perceived validity; however, involving two phases of diverse experts increases the potential relevance of the framework to other contexts.

## INTRODUCTION

The COVID-19 pandemic has changed our world, having affected every sector significantly, including health, education, economic, social, cultural and informational. One of the social repercussions of the pandemic has been the constant spread through various media of overwhelming volumes of information, particularly concerning health, public health, government directives and related issues. Much of this has been 'misinformation' and 'disinformation', both of which refer to incorrect or misleading content, the difference being the intentionality of those engaging in disinformation to cause harm, whereas misinformation is non-malicious but still potentially dangerous. Sources of misinformation and disinformation range from non-stringently reviewed rapid academic

publications with non-credible or flawed methodologies—and thereby dubious conclusions (misinformation), to 'fake news' through anonymous social media posts and intentionally misleading messaging by government officials (disinformation).[1]

False information, combining accidental and intentional, has contributed significantly to misguided health policies and to a host of deleterious consequences for individual and population health.[2 3] This phenomenon is called an 'infodemic'. The WHO defines an infodemic as, 'the widespread distribution of false or misleading information in digital and physical environments during a disease outbreak'.[4] Without robust systemic safeguards in place, an infodemic can make communities, jurisdictions and whole populations more vulnerable to disease infection and their side effects, as well as to other related harms.[5] Information overload, including the infodemic, during the COVID-19 pandemic, has represented a parallel pandemic whose transmission rate is much faster than the disease itself, since rampant erroneous and prejudicial information can trigger the spread of wild and accelerated waves of fear and defiance in the general population.[6] In Iran, for example, there is evidence, though limited, that the infodemic that spread widely through social media during the pandemic was associated with significant COVID-19 vaccine hesitancy rates,[7 8] substantial uptake of traditional and complementary medicine products,[9] and poor adherence to preventive measures, such as masking, in the general population.[10] This escalation reinforces the importance of infodemic management in Iran.

Infodemics can severely change a pandemic's course by undermining public health and government recommendations and by diminishing population and community adherence to public health interventions such as masking, social distancing and vaccination. Economically disadvantaged countries are at higher risk of infodemics than developed countries, due to a range of inequalities.[11] Lower rates of health literacy, limited access to reliable health information and minimal public trust in public health authorities[11] can make people from underdeveloped and developing countries more susceptible to fake news and misinformation.[12 13] This vulnerability is compounded by further inequalities in terms of comparatively limited healthcare infrastructure and reduced access to healthcare facilities and public health professionals, which make people from these countries more prone to sporadic and ill-advised health and public health behaviours.[14] In this context, the infodemic can pose a greater threat to populations in underdeveloped and developing countries during epidemics and health emergencies[15] by negatively influencing public risk perceptions and by undermining evidence-based policy creation and national and regional emergency responses.[16–18] These hindrances can increase the spread and burden of the pandemic and widen global health disparities.

Infodemics have become a global phenomenon, impacting citizens in every country.[19 20] Addressing them is a new challenge and priority in managing and responding to epidemics and health emergencies. To understand and counter the rapidly changing nature of the COVID-19 infodemic and to mitigate its negative effects, such as the further spread of misinformation, several novel strategies and initiatives have been established across public health settings globally. The WHO has been widely respected for developing highly credible guidelines and initiatives to combat misinformation and infodemic management across the world.[21] From early in the COVID-19 response, the WHO began to develop international strategies for infodemic management, in cooperation with other organisations, including the US Center for Disease Control and Prevention (CDC) and the Africa CDC. To track and address misinformation surrounding COVID-19 and HIV, the Joint United Nations Programme on HIV/AIDS and the Africa CDC have been operating a rumour management system—software that uses machine learning, combined with human expertise, to collect and analyse rumour data from open-source traditional media (web-based, news broadcasts), as well as social media (Facebook, Twitter, WhatsApp). The system enables the identification of false and misleading information related to COVID-19 and HIV.[21] In addition, the WHO developed a framework for infodemic management through crowdsourcing and online consultation with a wide range of global public health professionals.[22] Multiple countries like Ghana have taken steps to identify, analyse and respond to COVID-19 and vaccine-related misinformation.[23] These initiatives are helpful foundations for further infodemic management strategies.

Risk communication and infodemic management (RCIM) are the core of risk management and effective responses to epidemics and health emergencies.[24] According to Eysenbach, there are four pillars of infodemic management: information monitoring, building health and e-health literacy in the general population, consolidating and disseminating credible information, including by accelerating the academic peer-review process, to ensure accurate and timely knowledge translation, and minimising factors, such as political or commercial agendas, that can distort or distract from evidence-based guidance or strategies.[25] Combatting misinformation or disinformation for populations is as critical as ensuring much-needed medical equipment and supplies for health workers are readily available.[26] In underdeveloped and developing countries, given their existing health information inequalities and public health vulnerabilities, customised RCIM approaches are needed to combat infodemics and to reduce their effects on population health.[27] In particular, engagement and collaboration with local communities and leaders and stricter public health regulations are necessary.[27] While some contexts may be more susceptible to the dangerous potential impacts of misinformation and disinformation, none is immune, and the consequences of failing to tackle it directly and strategically can be dire.

The purpose of this study was to build on and extend previous conceptualisations of RCIM capacity-building by creating a conceptual framework of RCIM in Iran. To achieve this, we applied a systems thinking lens, since the pandemic demonstrated that not only can health emergencies affect all people and sectors, but that addressing infodemics requires more than just public health messaging. Along with potential benefits for other sectors, robust national and regional RCIM approaches can have a significant positive impact on health systems, those who bear the brunt of health emergencies. The WHO describes a health system as a set of interconnected building blocks that are essential to health system functioning. The blocks are: service delivery, health workforce, health information systems, access to essential medicines, financing and leadership/governance, with the latter being central to all. It is essential that each of these interconnected elements is addressed concomitantly in response to changing population health needs and inequalities, and to epidemics and health emergencies.[28] This multifaceted understanding of health systems, along with considerations for other related sectors, is vital to effective RCIM strategies, since misinformation and disinformation can affect those in all aspects of society. The nature of health emergencies requires that policy and communications strategy recommendations should be gathered from a diverse group of actors with relevant RCIM expertise, including researchers, educators, advocates, practitioners, funders, private sector representatives, community representatives, government officials, policymakers, and various trusted international experts and representatives. Leaders from across sectors should also collaborate with public health and with each other to integrate RCIM strategies effectively to improve the health of all people and communities.[29 30] Applying these diverse perspectives and the systems thinking approach can enhance RCIM policies, strategies and activities nationally, regionally, and locally and can lead to improved relevant health outcomes during epidemics and health emergencies.[31]

## METHODS

This sequential, mixed-methods exploratory study was conducted in two phases from October to December 2022. Phase 1 involved semistructured interviews with key informants from provincial and national public health authorities to inform the creation of an initial framework of key RCIM components across settings. Phase 2 involved an online panel of experts from relevant scientific domains to validate the conceptual framework's validity, credibility and transformability.[32 33] We then revised the framework based on the panel's feedback (figure 1). This study followed the Standards for Reporting Qualitative Research checklist.[34]

### Phase 1: semistructured interviews

Phase 1 involved semistructured interviews with a purposive sample of 20 Iranian public health professionals across provincial and national health authorities. Study participants included stakeholders, academics, decision-makers and leaders with expertise in community health, epidemiology, public health, social medicine, health communication and sociology. Participants were from eight prespecified provinces: Kerman, Tehran, Fars, Isfahan, Mazandaran, West Azerbaijan, Kermanshah, and Sistan and Baluchestan. These provinces were initially selected to involve a representative sample of the Iranian population with diverse social, geographical and cultural characteristics. Inclusion criteria were: (1) having at least 1 year of experience in either COVID-19 prevention and control programmes or decision-making in provincial or national public health settings, and (2) willingness to participate in the study.

An interview guide was developed according to previous studies (online supplemental appendix 1). The interview guide focused on the processes, infrastructures, challenges encountered and best practices relevant to RCIM during the COVID-19 pandemic in Iran. The interview guide was assessed beforehand by two expert reviewers. It was subsequently pretested with three target population members before the implementation.

The interviews followed a semistructured design, allowing for variations of the order of the questions and follow-up questions based on participant responses. The objectives and the activities that were involved in the study were explained to the participants. The principal investigator's contact details were provided, and participants' confidentiality was guaranteed. Written consent was sought before the interview, and the participants were asked to email the completed form to the principal investigator (online supplemental appendix 2). An experienced interviewer with a background in qualitative research and interviewing expertise conducted the interviews in the Farsi language. Due to COVID-19 social distancing, all interviews were conducted by telephone, audio-recorded and transcribed verbatim. Interviews ranged between 20 and 55 min (mean=34 min). Interviews lasted until the researchers realised they had reached content saturation.

To analyse the interview data, all interviews were transcribed verbatim. Then one of coauthors extracted concepts and open codes using Braun and Clarke's framework for thematic analysis of qualitative data[35] to the interview transcripts. The authors define thematic analysis as, 'the process of identifying patterns or themes within qualitative data' (p. 78). Their framework involves six steps: becoming familiar with the data, generating initial codes, searching for themes, reviewing themes, defining themes and writing up.

The initial set of open codes, themes and subthemes was discussed by participants and subsequently reviewed by the entire research team to improve the credibility and trustworthiness of the qualitative study. We used MAXQDA V.12 (VERBI, USA) for manual coding and content analysis.

| Leadership & governance | Information | Health workforce | Financing | Media | Community |
|---|---|---|---|---|---|
| **Structure**<br>Clearly defined personnel, roles, protocols, supports, and accountabilities<br><br>**Ethics**<br>Integrity, equity, transparency, and accountability<br><br>**System capacity**<br>Develop infrastructures for monitoring, social listening, communicating, and distilling the best available information and recommendations<br><br>**Operationalisation**<br>Develop, implement, and evaluate communication policies and strategies for potential risks<br><br>**Engagement**<br>Create population, multisector, and community involvement for risk communication<br><br>**Institutional/provincial design**<br>Polycentric governance to share information and inform and guide local responses, including by using crowdsourcing | **Platform**<br>Develop a networked platform for real-time and reliable data collection, including from international sources<br><br>**Surveillance**<br>Use the information to investigate, prepare, and effectively respond to challenges<br>**Policy making**<br>Use of surveillance data for evidence-based decision-making and recommendations<br><br>**Evaluation and adaptation**<br>Regular monitoring and analysis of the system for continuous quality and reliability of data sources<br>**Knowledge sharing**<br>Develop, implement, support, and facilitate accurate knowledge sharing | **Diversity and flexibility**<br>Prepare diverse specialists in all parts of the system with the various skills needed to respond effectively to evolving challenges<br><br>**Limited resources**<br>Prepare an adequate number of professionals, despite scarce resources<br><br>**Community involvement**<br>Resource capacity beyond health professionals, such as scientists and experts, professional councils, and NGOs. | **Resource allocation**<br>Ensure there is adequate financing and use resources effectively | **Mass Media**<br>Develop communication channels through Television, radio, and newspapers<br><br>**Government and health authorities' websites**<br>Develop validated sources of information with high accessibility and timelines<br><br>**Social media platforms**<br>Use social media to increase access to credible information and recommendations and to combat misinformation<br><br>**Source Credibility**<br>Build and maintain trust in formal sources<br><br>**Formal spokespersons**<br>Identify and prepare credible and qualified individuals | **Diversity**<br>Customise responses based on the diverse needs of communities with different cultures, ethnicity, and geographies<br><br>**Engagement**<br>Involve, consult, inform, engage, and collaborate with diverse communities and leaders<br><br>**Resilience**<br>Improve health, media, and digital literacy, and provide evidence-based tools<br><br>**Empowerment**<br>Enable communities to develop their own solutions and identify local influencers<br><br>**Trust**<br>Build and maintain trust to maximise social cohesion and successful response |

**Figure 1** A conceptual model of components and infrastructures of health risk communication and infodemic management system in Iran. NGOs, non-governmental organisations.

Based on the themes identified from the qualitative interviews and subsequent inspections, we created an initial set of 33 key RCIM strategies and organised them according to four of the pillars of the WHO model of the health systems, along with media and community.[28] This initial set of components served as the basis for discussion with, and validation by, the expert panel in phase 2 and consequently, the conceptual model.

### Phase 2: expert panel validation

Phase 2 involved a group of nine experts selected through purposive sampling to validate and prioritise key components of the initial RCIM model and to evaluate its completeness and validity.[32 33] The panel included a diverse set of stakeholders, academics, decision-makers, leaders from the various communities and national public health leaders. The inclusion criteria for this phase were: (1) having at least 3 years of professional experience or established research expertise in the fields of public health, epidemiology, crisis management, infodemiology, social media studies or health communication; and (2) willingness to participate in the study. Potential panel members (n=9) were identified through their academic or professional roles in health risk communication or risk management activities across provincial or national health authorities during the COVID-19 pandemic. Prospective contributors were given a short statement of the study's purpose and design and were invited by email to participate in the panel discussion. During the discussion, panellists engaged based on their assessments of the initial conceptual model and suggested additions, deletions and modifications, with the aim of informing a highly complete and credible model of essential components of an RCIM model for the country. As mentioned previously, this validation by experts was also intended to augment the quality, reliability and validity of the model.[32 33]

Following this phase, several revisions were made to the original conceptual model, but no factor was deemed required for exclusion. The required level of consensus for each component in this phase was a minimum of 75% agreement.

### Patient and public involvement

No patients or community members were involved in this study.

**Table 1** Demographic characteristics of the participants in the interviews (phase 1)

| Demographic characteristics | Frequency (%) |
|---|---|
| Residence at the time of interviews | |
| Tehran | 7 (35) |
| Kerman | 5 (25) |
| Fars | 2 (10) |
| Isfahan | 2 (10) |
| Kermanshah | 1 (5) |
| Mazandaran | 1 (5) |
| Sistan and Baluchestan | 1 (5) |
| West Azerbaijan | 1 (5) |
| Age | |
| 40–49 | 6 (30) |
| 50–59 | 11 (55) |
| ≥60 | 3 (15) |
| Gender | |
| Men | 19 (95) |
| Women | 1 (5) |
| Academic discipline | |
| Epidemiology | 6 (30) |
| General medicine | 4 (20) |
| Sociology | 3 (15) |
| Health policy | 2 (10) |
| Infectious disease | 2 (10) |
| Social medicine | 2 (10) |
| Health education and promotion | 1 (5) |

## RESULTS
### Phase 1: semistructured interviews
#### Participants

Most participants were men (n=19, 95%), aged 51–60 years old (n=11, 55%), from medical and public health disciplines (n=17, 85%), who work as a provincial or national health officer (n=14, 70%). Participants were mostly from Tehran (n=7, 35%) and Kerman provinces (n=5, 25%) (table 1).

The analysis of the qualitative data collected during the key informant interviews revealed 948 open codes and 84 subthemes. Subthemes were subsequently classified into 33 components (online supplemental appendix 3).

The next step involved organising these components according to six categories representing a combination of the WHO model[24] and key aspects of the Iranian health system: leadership and governance, information, health workforce, financial resources, media and community. The results formed the initial RCIM conceptual model.

### Theme 1: leadership and governance

Leadership and governance are at the heart of the WHO model of health systems[28] and Dr Tedros Ghebreyesus, Director-General of the WHO, said in the early months of the pandemic, "The greatest threat we face now is not the virus itself, it's the lack of global solidarity and global leadership."[36] Similarly, in an international study of crisis leadership featuring 32 coauthors from 17 countries, Geerts et al highlighted that effective leadership, trust in leaders through transparent decision-making, communication and accountability are vital to successful public health strategies.[32 33]

These examples reinforce the finding in our study that every respondent mentioned leadership and governance as essential foundations for the RCIM model. Seven respondents emphasised transparency in decision-making, effective communication and accountability as important characteristics of effective leadership and governance. According to these respondents, a lack of transparency and accountability among Iranian health officials and government authorities was among the country's substantial weaknesses in risk communication and had adverse consequences. Thirteen respondents suggested that senior public health officials intentionally caused non-transparent information communication during the COVID-19 pandemic, motivated by financial and other competing interests, which, they suggested, eroded public trust significantly. Similarly, regarding sources of false messaging, seven respondents indicated that pharmaceutical companies were a major source of spreading misinformation during the pandemic. These respondents suggested that public health officials should allow their financial and competing interests, including those related to pharmaceutical companies, to deter them from spreading credible information about the efficacy of some new and underdeveloped medications and vaccines. One respondent expanded a perception that many health officials were among shareholders of the pharmaceutical industry, they advertised some drugs or public health products and subsequently caused a fake and unrealistic demand among the population.

Almost all respondents (n=18) emphasised that the health system needs a robust risk communication strategy and increased infodemic management capacity by developing infrastructures for monitoring the public's risk perception, knowledge and attitudes, communicating with the public and providing clear guidance through various media based on the best available science. Increased RCIM capacity would enable early detection of outbreaks of potentially harmful misinformation and disinformation, and quick responses to counter falsehoods with facts or other reliable information in a targeted way for each audience. One respondent suggested that building capacity should involve designing an infodemic management system that defines national and provincial responsibilities based on lessons learnt from credible global guidelines, national and regional successful strategies, challenges and failures, as well as leading practices, locally and elsewhere. This respondent added that the system should include a national independent core rapid response team with clear roles, protocols and accountability to collaborate with communities to screen

and identify their needs, concerns and misinformation sources, to lead quick responses to the potential risks, and to prevent or mitigate the viral spread of misinformation and disinformation across the communities. Similarly, six respondents proposed developing, implementing, evaluating and revising communication policies and strategies to confront potential risks. According to these respondents, the lack of national and provincial policies and programmes for RCIM severely inhibited the national COVID-19 control and management efforts. Four respondents argued that the government's poor management of the COVID-19 infodemic, poor communication with the public and other stakeholders, and a lack of national and provincial strategies to address misinformation were major shortcomings of RCIM in Iran.

To optimise RCIM strategies, it is crucial to involve representatives from multiple sectors and the community representatives. Eleven respondents indicated that top-down public health initiatives that lacked community-based customisation and approaches were among the major barriers to acceptance of COVID-19 prevention and control interventions during the pandemic. Multi-sector and community involvement could also potentially improve community members' motivation to participate actively in information communication and management of infodemics. For example, one respondent described how social influencers in community-based approaches, such as that in Safiran-e-Salamat, Tehran, served as facilitators for effective RCIM across provincial settings.

Ineffective use of institutional and provincial infrastructures and capacities and lack of crowdsourcing were cited by four participants as major barriers to effective infodemic management during the COVID-19 pandemic. These respondents elaborated that medical universities and faculty within the provinces were isolated from the national health authorities and not supported by the Ministry of Health in planning and decision-making. These two respondents recommended establishing official networks of experts in diverse areas and practitioners to share experiences, challenges and best practices of information communication during the potential risks and increase capacity.

Along with trustworthy public health guidance and recommendations, two respondents stated that providing all people with tools for filtering, assessing and fact-checking information is essential to combat misinformation during the pandemic and health emergencies. Five respondents believed that using a well-known and reliable communication channel and technology-based interventions would maximise the spread of valid information and impact communication efforts and strategies.

## Theme 2: information
Developing a network platform to systematically collect, analyse and interpret epidemiological data from the community and quickly disseminate the key findings was considered an important characteristic of risk communication by 14 respondents. These respondents emphasised

that a lack of access to real-time, valid and high-quality data about the incidence, mortality and burden of COVID-19 in different provinces intensified the potential risk and spread of misinformation among the population.

Similarly, seven respondents indicated that a lack of access to high-quality surveillance data for research activities and to inform responses to potential and emergent challenges reduced the reliability of information and recommendations and transparency of government decisions. Consequently, it raised dramatic social concerns about the government's ability to estimate the spread of the disease and to anticipate and evaluate the effect of specific policies on population health.

In addition to data quality issues, two respondents suggested that the lack of substantial resources to handle the multiplication of data sources and information producers, to monitor disease trends regularly and to appraise the quality of data sources was a major barrier to the effective use of surveillance data for decision-making during the pandemic.

One respondent stated that some politicians, health officials and media misinterpreted and selectively reported data according to their own financial, commercial and political interests, which he considered a major source of misinformation during the pandemic. Two respondents argued that effective knowledge translation of high-quality data is required to minimise the spread of misinformation across different sectors and communities, since people's political, commercial and financial interests can lead them to distort scientific messages.

Finally, three respondents reported that these data issues contributed to a lack of evidence-based policies and practices, which severely inhibited effective RCIM.

## Theme 3: health workforce
All respondents highlighted the need for well-trained specialists in various organisations with a mix of skills that can contribute to RCIM activities, as well as additional training for all health workers.

Six respondents promoted the benefits of involving public health agencies, epidemiologists, data scientists and sociologists who have unique expertise and credibility to guide policies, strategies and RCIM, in collaboration with health workers. However, all respondents agreed that the Iranian scientists and experts have not helped substantially to prevent misinformation and to mitigate the effects of the infodemic. Further, three respondents suggested that, in some cases, scientists and academic experts in infodemic management were considered sources of misinformation, seen as contributing to the infodemic trends by publishing low-quality scientific papers and providing non-credible, sensational or exaggerated information about new treatments.

To gather relevant data and to disseminate evidence-based guidance, 12 respondents highlighted the need to involve professional councils, non-governmental organisations (NGOs), interested experts and health volunteers

as additional workforce sources to improve the speed and effectiveness of the response to the infodemic.

Three respondents identified a key gap in RCIM: a lack of qualified and well-trained spokespersons in public health and health organisations, which, they suggest, greatly diminished the quality of risk communication activities during the pandemic. Four respondents referenced a range of competencies necessary to improve the quality of the health workforce education and practice, which can be used to select potential candidates for RCIM roles and to design educational courses and curricula to enhance their ability to support health emergency response effectively.

### Theme 4: financing

Three respondents highlighted the importance of effective financial resource allocation to RCIM to support data collection and analysis and communication strategies. One respondent argued that multisector collaborations could reduce the risk of underfunding communication responses. Using technology-based interventions, such as text-messaging approaches, could improve the cost-effectiveness of communication strategies.

### Theme 5: media

All respondents mentioned characteristics related to media. Two respondents believed that given the broad coverage and penetration of radio and television (TV) as dominant communication channels in most parts of the country, involvement of trustworthy spokespersons in, and collaboration with, mass media, could improve the effectiveness of risk communication strategies. However, the respondents elaborated that the weak contribution of these media in RCIM was an obstacle to preventing misinformation. Even worse, nine respondents argued that TV and other mass media actually contributed to the COVID-19 infodemic. According to these respondents, broadcasting news reports that included misleading and low-value information, interviews with non-experts, and flagrant criticisms or debates about the performance of public health agencies reduced public trust and prompted many people and communities to rely more on informal and social media channels.

Three respondents added the need to improve government and health authorities' websites to disseminate real-time and high-quality information, since many consider them the source of credible information. Additionally, six respondents advocated social media platforms as important communication channels for most communities to aid the acceptance of public health interventions. Three respondents elaborated that reduced public trust in formal and government communication channels caused many people to rely instead on social media platforms, viewing them as more trustworthy. For example, according to two respondents, the dissemination of valid and high-quality data through social media channels influenced the impact of local interventions and improved vaccination coverage for vulnerable and ethnic populations, particularly in Sistan and Baluchestan and West Azerbaijan. These respondents explained that, due to higher accessibility, social media platforms were highly used by younger adults and geographically distanced locations and, therefore, effective in improving the speed and effectiveness of interventions among members of these populations.

### Theme 6: community

Eight respondents reinforced the importance of involving the community in RCIM in two ways. First, by understanding their diverse demographic, social, economic and cultural compositions and by identifying their information needs, preferred media and key influencers. Second, by listening to their concerns, sharing key data and evidence-based recommendations with them, and incorporating their input transparently into important, relevant decisions. However, four respondents suggested that the lack of community-centred approaches reduced the effectiveness of risk communication efforts during the COVID-19 pandemic in most Iranian provincial settings. Three respondents recommended priority training in critical thinking, media and health literacy for community leaders in RCIM to improve their engagement, active contribution and effectiveness. According to these respondents, well-informed, engaged and enabled communities can minimise misinformation and infodemic consequences and develop their own local solutions. One respondent expressed that this kind of respectful, reciprocal relationship with communities could rebuild and maintain public trust in public health agencies, health professionals and government authorities and could also maximise social cohesion and local capacity successfully to respond to potential risks during the crisis.

### Phase 2: expert panel validation

In this phase, the completeness and trustworthiness of the proposed conceptual model of RCIM in Iran were discussed by the online expert panel until consensus was achieved by all panel members (100% agreement) (figure 1).

## DISCUSSION

This study, conducted during the COVID-19 pandemic, was inspired by an awareness of two aspects of the global experience. The first is the extent to which infodemics can influence the course of large-scale health emergencies, given the global impact that the COVID-19 infodemic has had on individual and population health.[2 3] The term 'infodemic' refers to the profusion of recurring waves of information of overwhelming volume and predominantly unclear and/or mixed credibility, including disinformation, messaging intended to deceive. Infodemics can erode the quality and effectiveness of policy and strategy decisions. They can also intensify community and population-level distrust in government and public health officials and experts, including their recommendations,

which can drastically undermine national and local efforts to effectively mitigate the spread of the disease. As people's faith in official sources diminishes, the likelihood of them being influenced by alternatives increases, and the escalation of rumours and fear exacerbates. Broadcasts of incorrect information through TV, radio, newspapers, and other mainstream and social media, and even through academic publications, can contribute to widespread non-adherence to public health directives, thereby perpetuating the spread, impact and burden of a pandemic.

Infodemics can have increasingly devastating effects in economically disadvantaged countries, due to a wide range of inequalities,[11] which can make local populations more susceptible to fake news and misinformation.[12][13] This vulnerability is compounded by further inequalities in terms of healthcare infrastructure, access to healthcare facilities and health professionals.[14] Evidence suggests that, in Iran, the infodemic spread, largely through social media, contributed to several adverse outcomes in the general population.[32] The speed, scale and potential lethal consequences of infodemics are why they are considered parallel pandemics, which require a dedicated, strategic, expertise-informed response to allay.

The second inspiration for the study was an appreciation for the vital mitigating role that effective RCIM can play in pandemic and infodemic response. Understanding the sources of misinformation and disinformation and rapid, effective government and public health response, in collaboration with multisector and community leaders,

to evolving risks, along with targeted strategies, can mitigate potential negative ramifications.

The purpose of this research was to support increased national and local RCIM capacity in Iran and beyond by creating a unique conceptual model of evidence-informed, expert-informed and experience-informed strategies for RCIM during epidemics and health emergencies. To create the model, we applied a systems thinking lens, since infodemics and their effects reside within multisectoral complex systems involving interactions and actors from all aspects of society. This perspective considers how to most effectively engage with potential audiences and diverse stakeholders, including the community, scientists and experts, government and public health officials, health workforce, pharmaceutical industries (private sector) and others, through physical and virtual communication channels (figure 2). This comprehensive approach can enhance the potential for sectoral and provincial health authorities to improve RCIM activities and relevant health outcomes during epidemics and health emergencies. Given this perspective, following leading international pandemic research,[27] we gathered two stages of input and validation from diverse groups of those with expertise and experience in public health and various related sectors and disciplines.

The model presented here is organised according to four pillars of the WHO model of the health systems, along with media and community,[28] and it is reinforced by the full consensus of an expert panel in terms of its quality, completeness and validity. While the model was

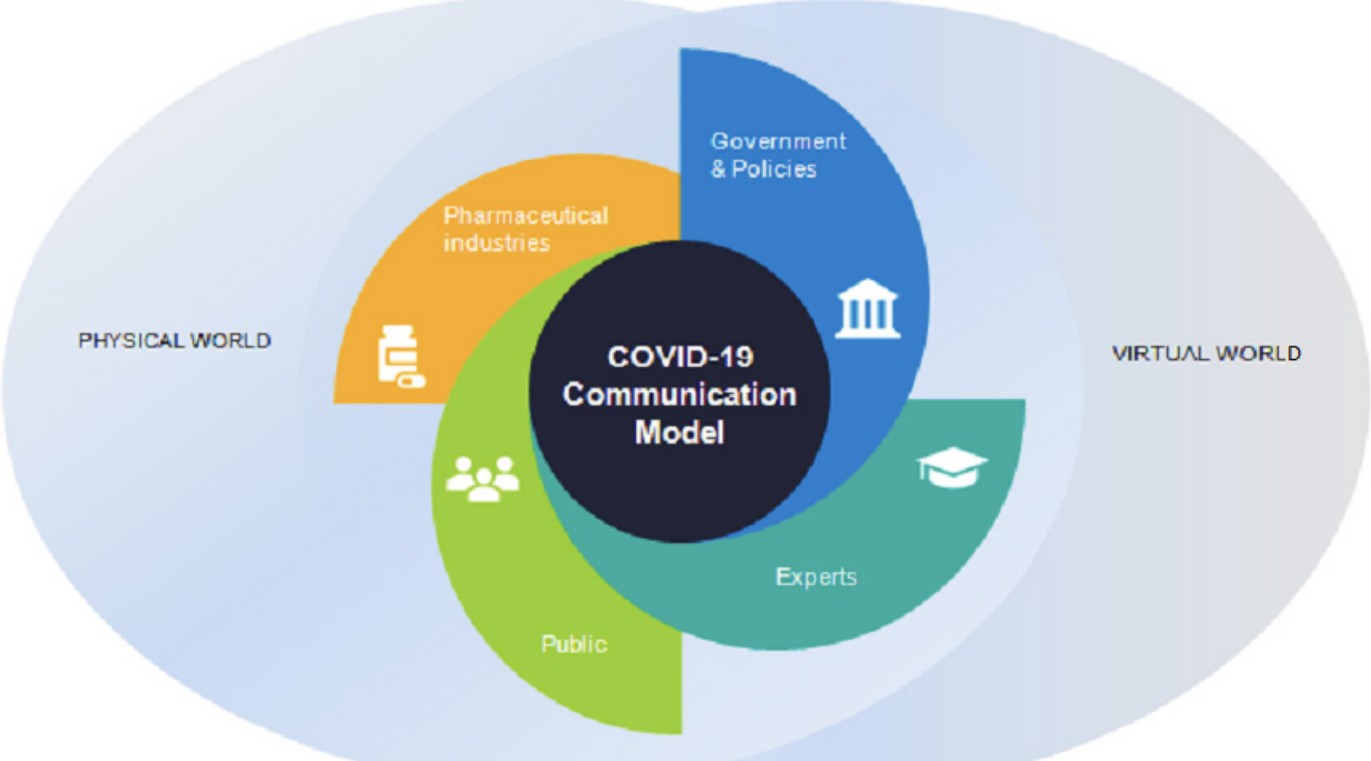

**Figure 2** Components of the COVID-19 risk communication and infodemic management in Iran (source: own production).

developed for the Iranian context, the intention was for it to have potential application in other contexts to decrease the spread and burden of future health emergencies and to minimise global health disparities.

What follows are some insights on, and priority points for, effective RCIM that emerged from the qualitative interviews and expert panel discussions.

Our findings support the vital importance and potential impact of establishing a robust, integrated, evidence-informed national RCIM strategy, with regional applications, to strengthen existing RCIM capacities to explore, track, monitor, respond and adapt to the needs of each community. Our results also show that effective RCIM requires several essential components: an official RCIM strategy supported by dedicated personnel, infrastructure, financing and resources, trustworthy leadership and governance, the expertise and capacity to inform policies and to gather, analyse and communicate the best available information in real time, effective messaging through mainstream and social media with local support, RCIM training for specialists among the healthcare workforce and community engagement to maximise local outcomes.

## Official RCIM strategy with dedicated personnel, infrastructure, financing and resources

Effective RCIM requires having an official strategy, based on a credible conceptual framework, which drove this study, and consolidated lessons learnt locally and elsewhere. Aspects of the strategy need to evolve and adapt based on changing circumstances and it is essential that consideration is given to roles and customised approaches at the national, regional and community levels. This should involve an official core national rapid response team with clear roles, protocols, resources and accountability, along with regional chapters.

Second, the strategy needs to be supported by the infrastructure, financing and resources to operate effectively. Respondents in our study suggested that in Iran, however, funding to enhance RCIM system capacity in terms of infrastructure and personnel is poor, and they indicated that the lack of direct funding hindered the risk communication support during the pandemic. Although there are media and public communication experts, the number of those available with expertise and training in responding to major health risks is critically limited. Underfunding RCIM appears to be a common challenge in many countries. Evidence from Southeast Asia,[37] for example, revealed that during the COVID-19 pandemic, few countries allocated resources to emergency risk communication. However, some specific areas have budgets, such as information education communication materials. Also, resource mobilisation and the use of non-governmental resources were reported as strategies to address this critical challenge within the country's national and provincial settings. Priority areas and optimal mobilisation and use of resources are important considerations for further exploration.

## Leadership and governance

The COVID-19 pandemic has highlighted the global importance of trustworthy and effective leaders who keep people at the forefront of their decisions, which they make transparently based on the best available evidence from a systems thinking perspective, and hold themselves accountable for outcomes.[27] Leadership and governance are also at the heart of the WHO model of health systems.[28] Similarly, every respondent in our study reinforced the fundamental importance to effective RCIM of leadership and governance.

Leadership-wise, effective RCIM response involves ensuring that the official RCIM strategy, personnel, infrastructure and resources identified in the previous point are in place. But these are insufficient on their own.

Effective RCIM leadership and governance depend on government officials and public health and other leaders earning people's trust through their integrity and public versus self-interest. If either of these are considered compromised, RCIM efforts are vastly undermined, as was seen during the pandemic in Iran. Leaders also earn trust by instilling confidence that, in a timely manner, they have the expertise and capacity to access and interpret the most credible information, operationalise an evidence-informed strategy and adapt it when necessary, and make and communicate transparent decisions, along with their rationale. Credible information should be actively gathered from many sources, including international, national and local experts, leaders in all related sectors, and community leaders and representatives. Effective leaders understand that tailored, two-way communication according to an accurate understanding of each stakeholder's and community's preferences is crucial. This communication involves asking important questions, active listening, sharing information, providing clear recommendations, tools and customised messaging, and engaging local support to lead RCIM. Finally, respondents indicated that leaders need to hold themselves publicly accountable for outcomes.

Leaders' ability to deliver on their responsibilities requires the aforementioned strategy, personnel, infrastructure and resources, as well as developing a network of diverse international, national and local experts in various relevant disciplines, leaders from all sectors, RCIM specialists within the health workforce and community leaders.

## Information

Effective RCIM relies on three approaches to information. The first is the expertise and capacity to, in a timely manner, proficiently screen, monitor and verify the validity, relevance and potential impact of available information from official and unofficial sources. The second is the ability to actively gather information from those with relevant expertise related to pandemic response and to RCIM strategies. The third is to communicate the most credible information to inform policymakers, government officials, public health, community leaders, and

health and healthcare practitioners to equip them with the knowledge to create, implement, and adapt appropriate and effective strategies.

## Media and communications

Combatting infodemics hinges on credible and strategic messaging through official sources, including government and public health websites, as well as through mainstream and social media, in collaboration with local representatives. The collaborative contribution of the government, public health, leaders in various sectors, experts and community leaders in circulating health information is a key strategy to counter misinformation or disinformation during health emergencies. Understanding the needs, perceptions, priorities and concerns of key stakeholders across public and private settings and identifying different opportunities and strategies for their involvement are critical steps to developing and implementing risk communication policies and strategies.

Developing or sustaining reputed and well-trusted communication channels is critically required to maximise the effectiveness and impact of communication strategies. How the community perceives various epidemics and health emergencies, what they perceive to be their role, how they are influenced and how their views tally with the biomedical approach are not entirely investigated in the country.

According to our findings, a lack of public trust in mass media and government channels directed Iranian citizens to the wide use of online social networks. Due to the dramatic reduction in social capita, most Iranians distrust governmental information sources, and this fact challenged the community's compliance with preventive behaviours (COVID-19 vaccination) during the COVID-19 pandemic. Lack of trust in the government as a source of information was reported globally in the existing literature. According to recent evidence, only 40% of the European citizens from the Economic Co-operation and Development countries participated in a survey and trusted their governments as sources of information about the coronavirus.[38] False claims about the activities, statistics, or policies of public and government authorities were reported as a major source of disinformation during the COVID-19 pandemic, suggesting that 'governments have not always succeeded in providing clear, useful and trusted information to address pressing public questions'.[39] Meanwhile, disinformation and claims may also be falsely attributed to official and governmental sources, amplifying this problem. In this regard, delivering truthful, evidence-informed and compelling information to various audiences through their preferred channels and understanding behavioural and psychological biases are recommended. This is especially important for young audiences, who tend to access news and information predominantly via social media platforms.[40] It is, therefore, a critical issue for health RCIM to ensure key factual messages reach all audiences. It is also important to effectively leverage the channel through which various audiences are relayed since different groups are likelier to trust media outlets that align with their views.

## RCIM training for health workforce

While some capacity-building workshops for health professionals were held during the COVID-19 pandemic by the Ministry of Health and medical universities, they were largely been ad hoc, of short duration (less than a week) and of variable quality. Those trained have often been public health professionals who then move on to other areas of public health. A planned and institutionalised approach to capacity-building is required to have an adequate pool of trained experts for epidemics and health emergencies. Therefore, financial resources and building risk communication expertise are critical priorities for the country. Obtaining both these resources will require the endorsement of senior policymakers. Advocacy to policymakers and key decision-makers on the role and impact of RCIM is very important.

## Training

RCIM is a broad and multidisciplinary field involving health communication, health education, public affairs, behaviour change communication and social mobilisation. It is therefore required to build the capacity of key contributors to verify, filter and curate health information and use diverse communication channels to target public audiences.[41] Community-based organisations, patient advocacy groups, professional associations and NGOs with reputable brands, organisational resources and a network of relationships can be leveraged to improve health risk communications. Existing evidence demonstrates that by partnering with local public health experts and policymakers to create information hubs and community outreach programmes,[42] these groups can significantly improve their ability to serve the information needs and concerns of diverse communities while also advocating for policy solutions. Existing evidence demonstrates that involving community members as planners, and attendees in pre-crisis planning activities, leads to increased preparedness and response activities. Therefore, training in roles and responsibilities, relationship building and team building are required strategies to facilitate and strengthen the contribution of community-based organisations, expert associations and other relevant partners during epidemics and health emergencies.[43]

## Community engagement

Effective RCIM depends on engaging with communities to share information and to understand their unique concerns, experiences, wisdom, available resources and preferred forms of communication, as well as to earn the support of community leaders as key intermediaries in response. These measures can maximise community collaboration and receptivity to ensuing recommendations. Given the social, contextual, economic and geographical diversity that exists within countries, customised, community-based approaches are essential

for RCIM and health emergency response. Ethnographic and anthropological/social research on epidemics and health emergencies in the country could also help to improve understanding of the acceptability of response to emergencies and public health interventions. According to our interviews and expert panel discussion, the community was considered a missing piece in RCIM strategies in Iran. Information needs and concerns (eg, disabilities, gender, age, literacy, cultural/ethnic backgrounds, access to technology) of the general Iranian population remained unexplored. In addition, the participatory engagement of citizens in a collective response to the COVID-19 infodemic was not only insufficient, but rather, at times, it was discouraged.

During the COVID-19 pandemic, national health authorities and governments in most countries predominantly demonstrated top-down communication strategies.[44] Effective RCIM requires a whole-of-society effort to sustain a healthy information ecosystem. Understanding the needs and concerns of vulnerable groups who might experience barriers to accessing accurate health information, care and support, or be at higher risk of exposure and secondary impacts, such as children and adults with disabilities, is critically important.[44] Effective risk communication can save lives during epidemics and health emergencies; however, existing evidence revealed that inadequate risk communication resulted in high exposure and loss of lives, as seen in Iran and Italy in the first wave.[44 45] Training and advising the general population on how to consume and share health information responsibly may be an effective strategy to improve the engagement and participation of public communities in RCIM. Investing in the community's media literacy, health literacy and critical thinking skills before the crisis can prepare society to mitigate the physical and emotional consequences of false news and disinformation and increase resilience.[46] As disinformation and infodemic during epidemics and health emergencies undermine trust, amplify fears and consequently affect countries' responses to the global pandemic, tailored strategies to build and maintain trust among the public community are of utmost importance. Therefore, to be effective and foster public trust in government, any activities conducted in health RCIM must be guided by the principles of transparency, integrity, accountability and community participation.

## Limitations

We address some limitations of the study. First, given that our study and the novel conceptual framework presented here are the first to address comprehensively the RCIM needs of, and strategies for, the Iranian health system context, further research and validation of its completeness and reliability, particularly after attempts to implement it, would be useful. Similarly, investigating causality and replicating the study with identical results can be challenging with qualitative studies of complex phenomena. However, involving diverse sets of respondents with

experience and expertise in leading RCIM in two phases of research before reaching total consensus heightens the potential for the framework to be considered credible and effective in being applied in the Iranian context. Further research could focus on applying best practices in RCIM, ecosystem mapping and analysis, and strengthening data collection and analysis for monitoring, evaluation, and learning. Investigating specific methods for evaluating RCIM activities is also important and critically recommended. Second, by focusing on the Iranian context, the transformability of the framework to other contexts remains yet untested. However, the high-level results echo leading international research on effective pandemic response and even if regional customisation would be beneficial, the current framework could potentially represent a well-informed basis for discussion, for further research and for the creation of local versions.

## Conclusion

This study was inspired by an appreciation for the extent to which the COVID-19 infodemic is reported to have impacted the spread and burden of the disease globally, and of the role that an effective RCIM strategy can play in mitigating the impact of infodemics. The purpose of this research was to support increased RCIM capacity in Iran and beyond through the creation of a unique conceptual model of evidence-informed, expert-informed and experience-informed strategies for RCIM during epidemics and health emergencies. Our findings suggest that ineffective RCIM impeded the emergency response in Iran's COVID-19 management, which is partly attributable to Iran's government and national public health authorities failing to infuse an evidence-informed and strategic RCIM into policymaking and decision-making. Consequently, access to high-quality and real-time information was extensively restricted and not publicly available, and the provincial public health settings failed to establish effective community relationships with experts, researchers, professional councils and NGOs to facilitate knowledge translation and utilisation. Further, the extensive use of social media platforms and mass media worsened the circulation of rumours, fake news and disinformation and led to public distrust. The lessons learnt from the outbreak management and response in Iran suggest that RCIM should be an essential component of health emergency readiness and response activities. This begins with trustworthy leaders at all levels who have integrity and make credible, transparent decisions, and hold themselves accountable for outcomes. A national RCIM programme should be established to support the required infrastructures, personnel and processes to address communication challenges during epidemics and health emergencies. This should be based on a conceptual model of RCIM to illustrate a collaborative and interdependent context of risk communication activities, implying that any improvements in these areas require an integrated and holistic approach. The government, private sector and pharmaceutical industries, experts

and the public should be involved in time, contributing diverse views and fulfilling respective responsibilities. The conceptual model presented here has the potential to be either implemented or serve as the foundation for the creation of a similar model in other contexts. Sharing experiences, challenges and leading practices among jurisdictions can further improve the reliability and credibility of guidance and strategies.

**Acknowledgements** The authors wish to thank the provincial and national public health professionals who contributed to this study, the WHO for their funding support, and the *BMJ Open* editor and peer reviewers for their valuable insights.

**Contributors** AB contributed to the project concept and manuscript design, qualitative data collection and interpretation, critical review of the manuscript writing and discussion of the manuscript. AS worked on data analysis, data interpretation and writing of the manuscript. MSB worked on data analysis, data interpretation and writing of the manuscript. HM worked on literature search, data interpretation and writing of the manuscript. MS worked on data analysis, data interpretation and writing of the manuscript. JG worked on data analysis, data interpretation, and writing and revising of the manuscript. HS worked on the project concept and manuscript design, supervising, critical review of the manuscript writing and discussion the manuscript. HS is responsible for the overall content as the guarantor. All authors read and approved the final manuscript.

**Funding** This study was financially supported by the WHO-Regional Office for the Eastern Mediterranean (WHO-EMR) Call for Proposals for Special Grant for COVID-19 Research, 2022 (WHO reference: 2022/1291032-0).

**Competing interests** None declared.

**Patient and public involvement** Patients and/or the public were not involved in the design, or conduct, or reporting, or dissemination plans of this research.

**Patient consent for publication** Not required.

**Ethics approval** This study was approved by the Research Ethics Committee of the Kerman University of Medical Sciences (IR.KMU.REC.1400.379). The Declaration of Helsinki was followed and informed consent was obtained from participants before starting the data collection stage.

**Provenance and peer review** Not commissioned; externally peer reviewed.

**Data availability statement** Data are available upon reasonable request.

**ORCID iDs**
Jaason Geerts http://orcid.org/0000-0001-6672-3859
Hamid Sharifi http://orcid.org/0000-0002-9008-7618

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
