## [Reviewer comments · BMJ Open]

ARTICLE DETAILS

TITLE (PROVISIONAL)	Health risk communication and infodemic management in Iran: Development and validation of a conceptual framework
AUTHORS	Bazrafshan, Azam; Sadeghi, Azadeh; Bazrafshan, Maliheh Sadat; Mirzaie, Hossein; Shafiee, Mehdi; Geerts, Jaason; Sharifi, Hamid

VERSION 1 – REVIEW

REVIEWER	Keating, Julie William S Middleton Memorial Veterans Hospital
REVIEW RETURNED	14-Feb-2023

GENERAL COMMENTS	Interesting work, the breakdown of infrastructure and needs as shown in figure 1 is very useful. I think some of the Iran-specific context in the discussion may be helpful to move to the introduction for readers - eg what negative effects were seen due to the infodemic? How did misinformation spread? This would provide context when reading the results. It's hard to tell if the results text is purely reporting what respondents stated or adding researcher editorial, which should be moved to the discussion. Consider using tables or graphical means to summarize the qualitative data from informant interviews. Also please add numbers - eg "more than half of respondents said" is how many? Recommend moving the risk communication model (figure 2) to the discussion and elaborate there on how these parties interact and the recommendations for supporting communication. Note there was odd spacing throughout, many lines were missing spaces between words. Not sure if this was in the original document or an issue with compiling into the PDF/packet. Figure 1 - Typo in "Leadership & governance" column ("ationsOperationali") Figure 2 - very pixelated to the point I can't fully read the text.
--

REVIEWER	Singh, Rita Hong Kong Baptist University, Language Centre
REVIEW RETURNED	03-Mar-2023

GENERAL COMMENTS	An interesting study involving the development of a conceptual framework for health risk communication and infodemic management during public health emergencies. WHO's health system framework is mentioned on page 3 but could it be elaborated in the Introduction? This is because the RCIM that
--

	you have proposed draws on this framework as you indicated on page 3. You can also explain what new components you have added to this framework based on your findings in the Discussion section so as to highlight your study's contributions. In the Introduction, disinformation and misinformation are used. Give a clear distinction between them according to the literature - they are not the same and interchangeable. In the Discussion and Conclusion, these terms appear too, so they need to be differentiated. The authors should provide the intercoder reliability statistics for the coding of the themes of the interviews. Braun and Clarke's thematic analysis needs to be explained further so that the reader can understand more about it. Thanks for including the detailed themes in the Appendix. Areas for further research can be added to the Discussion section. Minor issues: The manuscript needs to be proofread/edited. There are grammatical errors (e.g. line 13 on page 2 – constant release (not 'releases'); line 44 – professionals, making people...(not 'make people').
--	---

VERSION 1 – AUTHOR RESPONSE

Reviewer 1

Interesting work, the breakdown of infrastructure and needs as shown in figure 1 is very useful. I think some of the Iran-specific context in the discussion may be helpful to move to the introduction for readers - eg what negative effects were seen due to the infodemic? How did misinformation spread? This would provide context when reading the results.

Response: Thank you for your interest in the submission. We have inserted some information about the Iranian context and the impact of the infodemic on Iranian individuals in the introduction: "The COVID-19 pandemic and its accompanying infodemic have globally impacted individual and population health (2,3). In Iran, there is evidence, though limited, that the mis- and disinformation – the infodemic - spread widely through social media during the pandemic was associated with significant COVID-19 vaccine hesitancy rates (2, 3), substantial uptake of traditional and complementary medicine products (4), and poor adherence to preventive measures, such as masking, in the general population (5). This escalation reinforces the importance of infodemic management in Iran".

We have also added further points in the Discussion section.

"Evidence suggests that, in Iran, the infodemic spread, largely through social media, contributed to several adverse outcomes in the general population (32)".

It's hard to tell if the results text is purely reporting what respondents stated or adding researcher editorial, which should be moved to the discussion. Consider using tables or graphical means to summarize the qualitative data from informant interviews. Also please add numbers - eg "more than half of respondents said" is how many?

Response: Thank you for your consideration on this point. We have modified the qualitative results and added the number of respondents to each reference in the Results section.

Recommend moving the risk communication model (figure 2) to the discussion and elaborate there on how these parties interact and the recommendations for supporting communication.

Response: Thank you for your consideration on this point. We have moved Figure 2 to the Discussion section and elaborated on its essence and logic.

“To create the model, we applied a systems thinking lens, since infodemics and their effects reside within multi-sectoral complex systems involving interactions and actors from all aspects of society. This perspective considers how to most effectively engage with potential audiences and diverse stakeholders, including the community, scientists and experts, government and public health officials, health workforce, pharmaceutical industries (private sector), and others, through physical and virtual communication channels (Figure 2). This comprehensive approach can enhance the potential for sectoral and provincial health authorities to improve RCIM activities and relevant health outcomes during epidemics and health emergencies”.

Note there was odd spacing throughout, many lines were missing spaces between words. Not sure if this was in the original document or an issue with compiling into the PDF/packet.

Figure 1 - Typo in "Leadership & governance" column ("ationsOperationali")

Figure 2 - very pixelated to the point I can't fully read the text.

Response: Thank you for your consideration on this point. We have amended both figures to improve their readability and quality.

Reviewer 2

An interesting study involving the development of a conceptual framework for health risk communication and infodemic management during public health emergencies.

WHO's health system framework is mentioned on page 3 but could it be elaborated in the Introduction? This is because the RCIM that you have proposed draws on this framework as you indicated on page 3. You can also explain what new components you have added to this framework based on your findings in the Discussion section so as to highlight your study's contributions.

Response: Thank you for your interest in the submission. We have attempted to incorporate all the comments. We have inserted some information about the WHO framework used in this study in the Introduction section.

“The WHO describes a health system as a set of interconnected building blocks that are essential to health system functioning. The blocks are: service delivery, health workforce, health information systems, access to essential medicines, financing, and leadership/governance, with the latter being central to all (30)”.

Further, some explanation was added to link the study's framework to the Discussion section.

“The model presented here is organised according to four of the pillars of the WHO model of the health systems, along with media and community (30), and it is reinforced by the full consensus of an expert panel in terms of its quality, completeness, and validity”.

In the Introduction, disinformation and misinformation are used. Give a clear distinction between them according to the literature - they are not the same and interchangeable. In the Discussion and Conclusion, these terms appear too, so they need to be differentiated.

Response: Thank you for your consideration on this point. A simple definition of misinformation and disinformation was added to the Introduction section.

“Much of this has been “misinformation” and “disinformation”, both of which refer to incorrect or misleading content, the difference being the intentionality of those engaging in disinformation to cause harm, whereas misinformation is non-malicious but still potentially dangerous (1)”.

The authors should provide the intercoder reliability statistics for the coding of the themes of the interviews. Braun and Clarke’s thematic analysis needs to be explained further so that the reader can understand more about it. Thanks for including the detailed themes in the Appendix. Areas for further research can be added to the Discussion section.

Response: Thank you for your consideration on this point. As qualitative data were analyzed by one of our team in a short period of time, the consistency of coding was assumed to be favorable.

Therefore, no inter-coder or intra-coder reliability was measured. Information about the Braun and Clarke’s thematic analysis approach was added to the Methods section.

“To analyse the interview data, we applied Braun and Clarke’s framework for thematic analysis of qualitative data (37) to the interview transcripts. The authors define thematic analysis as, “the process of identifying patterns or themes within qualitative data” (p. 78). Their framework involves six steps: becoming familiar with the data, generating initial codes, searching for themes, reviewing themes, defining themes, and writing up”.

Some areas for further research were added to the Discussion and Limitations sections.

“First, given that our study and the novel conceptual framework presented here are the first to address comprehensively the RCIM needs of, and strategies for, the Iranian health system context, further research and validation of its completeness and reliability, particularly after attempts to implement it, would be useful”.

The manuscript needs to be proofread/edited. There are grammatical errors (e.g. line 13 on page 2 – constant release (not ‘releases’); line 44 – professionals, making people...(not ‘make people’).

Response: Thank you for your consideration on this point. We have conducted a thorough review of the manuscript. We have polished the writing, including by fixing grammatical and typographical errors in the previous submission.

VERSION 2 – REVIEW

REVIEWER	Keating, Julie William S Middleton Memorial Veterans Hospital
REVIEW RETURNED	19-May-2023

GENERAL COMMENTS	Nice paper, and thank you for the changes addressing the prior review. These have provided helpful context. I think the Introduction would benefit from editing for conciseness. It's very long and covers a lot of ground particularly with the added text, making it difficult to identify the points that are most relevant for this specific question and study. I wouldn't remove any major points, but just cut down a bit - for example, the paragraph on Infodemics (start page 3 line 42) could be edited down to mention that multiple countries are taking steps to combat misinformation and then cite examples for further reading, rather than detailing the programs in multiple countries. Methods - please add more specifics about the Phase I data analysis process on your team, i.e., how many people performed the initial analysis of reviewing transcripts and identifying themes, and if there was any larger team review of these themes and refining the codes in order to generate the 33 key RCIM strategies.
---

REVIEWER	Singh, Rita Hong Kong Baptist University, Language Centre
REVIEW RETURNED	06-Jun-2023

GENERAL COMMENTS	Thanks for addressing the comments and revising the manuscript.
---

VERSION 2 – AUTHOR RESPONSE

Reviewer 1

I think the Introduction would benefit from editing for conciseness. It's very long and covers a lot of ground particularly with the added text, making it difficult to identify the points that are most relevant for this specific question and study. I wouldn't remove any major points, but just cut down a bit - for example, the paragraph on Infodemics (start page 3 line 42) could be edited down to mention that multiple countries are taking steps to combat misinformation and then cite examples for further reading, rather than detailing the programs in multiple countries.

Methods - please add more specifics about the Phase I data analysis process on your team, i.e., how many people performed the initial analysis of reviewing transcripts and identifying themes, and if there was any larger team review of these themes and refining the codes in order to generate the 33 key RCIM strategies.

Response: Thank you for your interest in the submission. We have removed some information about the country specifics in the introduction and tried to reduce the word count from 1527 to 1261.

“The COVID-19 pandemic and its accompanying infodemic have globally impacted individual and population health. In Iran, there is evidence, though limited, that the mis- and disinformation – the infodemic - spread widely through social media during the pandemic was associated with significant COVID-19 vaccine hesitancy rates (2, 3), substantial uptake of traditional and complementary medicine products (4), and poor adherence to preventive measures, such as masking, in the general population (5). This escalation reinforces the importance of infodemic management in Iran”

Before the Internet, one of the main reasons for deaths during epidemics and pandemics was the lack of sufficient information on the prevention, care, and treatment of the disease (11). But as technology advances, during health emergencies, the profusion of information, which is often conflicting, increases, primarily through social and digital media and instant messaging (12). The potential consequences of this profusion can intensify, particularly when people are in lockdown or isolation (12).

“In particular, the impact of the infodemic on vaccination is critical because it is key to re-establishing pre-pandemic normalcy”.

“Drawing on the importance of this approach, we involved the perspectives of a diverse set of experts in our study to enhance the quality and reliability of the conceptual framework (34, 35). Our intention was for the framework to have the potential to be applied to build RCIM capacity effectively in Iran and in other underdeveloped and developing countries and beyond”

We have also rephrased your mentioned paragraph as following:

“Multiple countries like Ghana have taken steps to identify, analyse, and respond to COVID-19 and vaccine-related misinformation”

We have also added further points in the Method section.

“To analyse the interview data, all interviews were transcribed verbatim. Then one of co-authors extracted concepts and open codes using Braun and Clarke’s framework for thematic analysis of qualitative data (37) to the interview transcripts. The authors define thematic analysis as, “the process

of identifying patterns or themes within qualitative data” (p. 78). Their framework involves six steps: becoming familiar with the data, generating initial codes, searching for themes, reviewing themes, defining themes, and writing up.

The initial set of open codes, themes, and sub-themes was discussed by participants and subsequently reviewed by the entire research team to improve the credibility and trustworthiness of the qualitative study. We used MAXQDA 12 (VERBI GmbH, USA) for manual coding and content analysis.

Based on the themes identified from the qualitative interviews and subsequent inspections, we created an initial set of 33 key RCIM strategies and organized them according to four of the pillars of the WHO model of the health systems, along with media and community”.